# Highly Water-Stable Zinc Based Metal–Organic Framework: Antibacterial, Photocatalytic Degradation and Photoelectric Responses

**DOI:** 10.3390/molecules28186662

**Published:** 2023-09-16

**Authors:** Congying Yuan, Yadi Miao, Yinhang Chai, Xiaojun Zhang, Xiaojing Dong, Ying Zhao

**Affiliations:** 1College of Life Science, Luoyang Normal University, Luoyang 471934, China; pingboycy@163.com (C.Y.); myd17838906156@163.com (Y.M.); z2015797093@163.com (X.Z.); 13383799287@163.com (X.D.); 2Henan Province Function-Oriented Porous Materials Key Laboratory, College of Chemistry and Chemical Engineering, Luoyang Normal University, Luoyang 471934, China; chaiyinhangyh@163.com; 3College of Chemistry, Zhengzhou University, Zhengzhou 450001, China

**Keywords:** metal–organic framework, antibacterial ability, tetracycline degradation, photocurrent generation performance

## Abstract

A reported water-stable **Zn-MOF** ([Zn(L)_2_(bpa)(H_2_O)_2_]·2H_2_O, H_2_L = 5-(2-cyanophenoxy) isophthalic acid has been prepared via a low-cost, general and efficient hydrothermal method. It is worth noting the structural features of **Zn-MOF** which exhibit the unsaturated metal site and the main non-covalent interactions including O⋯H, N⋯H and π-π stacking interactions, which lead to strong antibacterial and good tetracycline degradation ability. The average diameter of the **Zn-MOF** inhibition zone against *Escherichia coli* and *Staphylococcus aureus* was 12.22 mm and 10.10 mm, respectively. Further, the water-stable **Zn-MOF** can be employed as the effective photocatalyst for the photodegradation of tetracycline, achieving results of 67% within 50 min, and it has good cyclic stability. In addition, the photodegradation mechanism was studied using UV-vis diffuse reflection spectroscopy (UV-VIS DRS) and valence-band X-ray photoelectron spectroscopy (VB-XPS) combined with the ESR profile of **Zn-MOF**, which suggest that ·O_2_^−^ is the main active species responsible for tetracycline photodegradation. Also, the photoelectric measurement results show that **Zn-MOF** has a good photocurrent generation performance under light. This provides us with a new perspective to investigate **Zn-MOF** materials as a suitable multifunctional platform for future environmental improvement applications.

## 1. Introduction

Pathogens have a high incidence rate and mortality, which directly threaten human health and the development of the ecological environment [1,2,3]. For various infections caused by bacteria, antibiotic therapy seems to be a long-term effective strategy. However, the widespread use of antibiotics has led to severe bacterial resistance and even the emergence of superbugs. Based on this current situation, the development of new antibacterial materials has become an urgent focus in the fields of materials science and biochemistry [4,5,6,7,8]. Among the previously reported antibiotics, tetracycline was widely used for the treatment of infectious diseases in humans and animals [9,10]. Non-biodegradability can easily result in health problems for aquatic organisms and increase the risk of antibiotic-resistant pathogens. Therefore, it is of great significance to develop stable antibacterial and antibiotic-degrading materials for the sustainable development of biology [11,12,13].

Metal–organic frameworks (MOFs) are an important class of compounds in which organic bridging ligands connect metal ions or metal-containing nodes to form a three-dimensional coordination network with potential voids [14,15]. The key advantage of MOFs over other microporous species is their highly adjustable composition [16], which can be achieved by using different metal ions or modifying organic linkers [17,18]. Due to their high surface area and open space, host–guest chemistry related to MOFs, such as energy-related technologies [19], heterogeneous catalysis [20], gas purification [21] and sensing [22], has been systemically studied. Because of the increasing number of multidrug-resistant bacteria in recent years and the residues of antibiotic drugs in the environment, there is increasing interest in developing new environmental applications of MOFs. In this regard, some transition metals [23,24,25] and metal nanoparticles (NPs) [26,27,28] have been extensively studied as antimicrobial agents and photocatalytic degraders. We also know that the degradable metal zinc has attracted the attention of researchers because it has a better degradation rate than magnesium and iron. Studies have shown that zinc has good biocompatibility in vivo, Zn^2+^ can promote cell growth and differentiation, and a high concentration of Zn^2+^ has certain antibacterial abilities. However, pure zinc does not display activity in vitro, and Zn^2+^ released by pure zinc is not enough to provide sufficient antibacterial activity [29]. Therefore, it has become an urgent problem to improve the surface bioactivity and cytocompatibility of zinc metal through the self-assembly of ligands and metals and to engender antibacterial properties and photocatalytic degradation properties.

Based on these questions, we used a case of water-stable Zn-MOF as previously reported by our research group [30] to study its antibacterial and photodegradation catalytic ability. Generally, molecular interactions and dynamics in the photocatalyst influence the activity, which in turn will be influenced by linker substitution. The aromatic carboxylic acid ligands can generate suitable electronic transitions, while self-assembly with zinc ions to form MOF can further improve stability and surface activity. Upon investigation, **Zn-MOF** has antibacterial properties and good photocatalytic degradation performance for tetracycline. The active substances easily produced by the catalyst were characterized by ESR and the mechanism of photocatalytic degradation was revealed. Finally, we also studied the photoelectric response performance.

## 2. Results

### 2.1. Antibacterial Properties of Zn-MOF

The results showed that **Zn-MOF** had inhibitory effects on both *Escherichia coli* and *Staphylococcus aureus*. The average diameter of the inhibitory zone of *Escherichia coli* was 12.22 mm, and that of *Staphylococcus aureus* was 10.10 mm, as shown in Figure 1 and Figure 2. The different antibacterial effects of the materials on *Escherichia coli* and *Staphylococcus aureus* may be due to the fact that the cell wall of *Staphylococcus aureus* (Gram-positive bacteria) is thicker than that of *Escherichia coli* (Gram-negative bacteria). This makes it more difficult for **Zn-MOF** to enter the cells of *Staphylococcus aureus*, thus preventing the bacterial structure from being destroyed.

### 2.2. Photocatalytic Properties of Zn-MOF

Tetracycline, an antibiotic widely found in the environment, was used to evaluate the photocatalytic activity of **Zn-MOF** in this work. First, the photocatalytic performance of **Zn-MOF** was tested, and the photolysis efficiency reached about 67% within 50 min (Figure 2a). Meanwhile, the change in the ultraviolet absorption spectrum of tetracycline during the photodegradation process is shown in Figure 2b. With the increase in irradiation time, the characteristic absorption peak of tetracycline gradually decreases, and the decrease in the absorption peak intensity indicates that most tetracycline molecules can be degraded and no other substances are formed.

However, in a blank control test, only negligible tetracycline degradation was observed in the absence of a photocatalyst (Figure 2a). In addition, kinetic characterization was performed, and the results showed that the photodegradation data fit the pseudo-first-order kinetic model. Figure 3 shows the dynamic curve, and the calculation formula −ln(*C*_0_/*C*t) = *kt*, where *k*, *C*_t_ and *C*_0_, respectively, represent the apparent rate constant of the reaction, the instantaneous concentration at time t and the initial concentration of tetracycline [31]. It is found that the degradation rate constant of **Zn-MOF** can be as high as 0.02627 min^−1^, while that of the control group is 0.0024 min^−1^, which represents a ten-fold increase. The results indicate that **Zn-MOF** has potential application value in removing tetracycline from an aqueous solution.

### 2.3. Cyclic Stability of Zn-MOF Photocatalysis

Cyclic stability is an important factor in evaluating the application of photocatalysts. Therefore, the stability performance of the photocatalyst was evaluated by comparing the degradation efficiency and the consistency of PXRD patterns before and after cyclic tests. After each cycle, the sample was cleaned with deionized water and dried. As shown in Figure 4a, the structure of the complex remained good after three cycles, and no obvious deactivation occurred. Meanwhile, in the third run, the degradation efficiency of tetracycline under visible light irradiation remained at 51.6% (Figure 4b), which indicated that **Zn-MOF** has good stability. At the same time, as shown in prior research [11,32], the phenomenon is basically consistent with that of relevant previous studies.

### 2.4. Analysis of Active Components in Photodegradation of Zn-MOF

In order to study the active substances produced in photodegradation, we characterized the materials via electron spin resonance (ESR) spectroscopy with the trapping agent 5,5-dimethyl-1-pyrroline N-oxide (DMPO) and further investigated their photocatalytic properties, which provided support for the catalytic mechanism. As shown in Figure 5, under dark conditions, it can be clearly observed that **Zn-MOF** does not have DMPO∙OH and DMPO∙O_2_^−^ signals. However, the typical DMPO∙OH (Figure 5a) and DMPO∙O_2_^−^ (Figure 5b) adducts were detected under visible light compared to the signals under dark conditions, indicating the successful formation of the two active substances. In addition, the signal strength of DMPO∙O_2_^−^ was significantly higher than that of DMPO∙OH when comparing the signal strength of the two active substances produced. These results indicate that ∙O_2_^−^ plays a key role in the photodegradation of tetracycline, while ∙OH plays a secondary role.

### 2.5. Morphology Measurement and BET Surface Area Analysis

In order to clearly show the morphological characteristics of **Zn-MOF**, Figure 6a displays its SEM image. The particles are relatively regular with a cubic structure, and the crystal has a uniform size on the micrometer scale. Furthermore, the pore structure, pore characteristics and specific surface area of the **Zn-MOF** sample were studied using adsorption and desorption techniques under 77 K nitrogen atmosphere. As shown in Figure 6b, the isotherm of **Zn-MOF** showed type IV, and hysteresis curves of the complex appeared under high pressure (*P*/*P*_0_), which proved that the pores in the structure were relatively concentrated [33]. Using the Barrett–Joyner–Halenda method, we found that the surface area and pore volume of **Zn-MOF** were 11.4087 m^2^/g and 44.64 × 10^−3^ cm^3^/g, respectively. The photocatalytic degradation can be related to different pathways. Although the small specific surface area and pore size do not provide positive factors for the photocatalytic process, the performance of **Zn-MOF** could be interpreted based on its textural properties. In addition to the direct photocatalytic degradation of tetracycline molecules absorbed by **Zn-MOF**, photogenerated electron−hole pairs can generate highly oxidative species such as superoxide anions and hydroxyl radicals. These mobile radicals will indirectly degrade the tetracycline in the surrounding solution. Furthermore, it is speculated that the main non-covalent interactions exhibited by **Zn-MOF** are O...H and N...H interactions, suggesting that tetracycline molecules can interact with materials through these interactions, which play a key role in the degradation of tetracycline molecules.

### 2.6. Possible Mechanism of Tetracycline Degradation

In order to more clearly analyze the production and action processes of hydroxyl radicals and superoxide radicals, we proposed a potential photodegradation mechanism based on the association between VB-XPS and UV-vis DRS analysis and ESR profiles. When illuminated by a photon with energy close to or higher than *E*g (**Zn-MOF**), electrons (e^−^) in the valence band (VB) will be excited and transferred to the conduction band (CB), ultimately causing the vacancy (h^+^) to remain in the valence band. As shown in Figure 7, the VB (2.76 eV) potential of **Zn-MOF** is more favorable than the VB potential of water molecules converting to hydroxyl radicals (2.38 eV vs. NHE) [34], resulting in visible light-generated holes (h^+^) reacting with water molecules to form hydroxyl radicals (·OH). At the same time, the CB potential of **Zn-MOF** (−1.15 eV) is more negative than the potential of oxygen conversion to superoxide free radicals (−0.33 eV vs. NHE) [35], which encourages the photoinduced electrons (e^−^) to react with oxygen to form superoxide free radicals (·O_2_^−^). However, from the mechanism diagram, we can also observe that the superoxide radical has a stronger production effect than the hydroxyl radical, which is consistent with the ESR spectrum.

### 2.7. Analysis of Photoelectric Response

The electrochemical workstation and the three-electrode system were jointly used to evaluate the photoelectric performance of **Zn-MOF**. According to the cyclic voltammetry curve (Figure 8a), it can be found that the working electrode has a strong photoelectric response to negative voltage, and its current density exceeds 20 μA cm^−2^ at a voltage of −0.6 V. In addition, under switching illumination (10 s) and a bias potential of 0.5 V, the working electrode exhibited a significant photoelectric response behavior (Figure 8c). Notably, periodic and persistent photocurrent signals are displayed in multiple on/off cycles. In order to further confirm the photoelectric response of the material, we also measured the photoelectric response at the opposite voltage −0.5 V bias voltage. Consistent with the cyclic voltammetry curve, the response is better at −0.5 V, which is about three times that under positive pressure (Figure 8d). In order to clarify the effect of additional light on the current of the working electrode, an electrochemical impedance spectroscopy (EIS) experiment was performed at a potential of −0.5 V (Figure 8b). Obviously, the addition of light can effectively reduce the charge transfer resistance.

## 3. Experiment

### 3.1. Synthesis of [Zn(L)_2_(bpa)(H_2_O)_2_]·2H_2_O

A mixture of Zn(OAc)_2_·2H_2_O (0.1 mmol, 20 mg), 5-(2-cyanophenoxy) isophthalic acid (H_2_L 0.1 mmol, 28 mg), 1,2-bis (4-pyridine)-ethane (bpa, 0.1 mmol, 16.8 mg) and H_2_O (8 mL) was sealed in a 25 mL Teflon-lined reactor at 120 °C for 3 days. When cooled to room temperature, regular crystals of **Zn-MOF** were obtained [30].

### 3.2. Materials and Equipment

The raw materials are all commercially purchased and do not require further purification. The solvents used are all analytical-grade solvents. Subsequently, the ESR was collected with Bruker A300 (Billerica, MA, USA), and the UV-vis diffuse reflectance spectrum was recorded with Shimadzu UV-2600 UV-vis spectrophotometer (Kyoto, Japan). Powder X-ray diffraction (PXRD) patterns were obtained using a Bruker D8 Advance powder diffractometer using Cu-Kα radiation (λ = 1.5418 A) at a scanning rate of 0.2 s/step and a step size of 0.02 (2θ) at 40 kV and 40 mA.

### 3.3. Antibacterial Performance Test

This work evaluates the antibacterial effect of **Zn-MOF** via the bacteriostatic zone method. Gram-negative bacteria (*Escherichia coli*) and Gram-positive bacteria (*Staphylococcus aureus*) were selected as research objects to measure **Zn-MOF** antibacterial performance. The diffusion of the sample in the solid culture substrate was used to inhibit the growth of bacteria and form a transparent circular bacteriostatic zone centered around it.

Firstly, the prepared **Zn-MOF** sample was ground and dispersed in dimethyl sulfoxide (DMSO), and 6 mm circular filter paper with a diameter of pre-sterilized bacteria was placed in the prepared **Zn-MOF** solution. Then, under aseptic conditions, the prepared LB (Luria–Bertani) solid medium plate was placed on the ultra-clean worktable next to the flame of the alcohol lamp. A total of 50 µL bacteria solution was absorbed and added to the LB solid medium. The bacteria solution was evenly coated with the sterilized coating stick, the filter paper soaked in LB liquid medium, and the filter paper soaked in **Zn-MOF** solution was removed with tweezers. Finally, the sealed medium plate was placed in a constant temperature incubator at 37 °C. After about 18 h of culture, it was taken out. A Vernier caliper was used to measure the diameter (in three directions) of the bacterial-inhibition zone centered on the sample, and its average value was taken as the experimental result.

### 3.4. Determination of Photocatalytic Activity

The photocatalytic performance of **Zn-MOF** on tetracycline was evaluated by using a 300 W xenon lamp equipped with a 420 nm filter to simulate the degradation of tetracycline. Firstly, the catalyst reached absorption–desorption equilibrium under dark conditions. The experimental method involved dispersing the ground photocatalyst (50 mg) in a tetracycline aqueous solution (20 ppm L^−1^) and stirring continuously for 60 min under dark conditions. Then, the reaction solution was exposed to visible light, and the reaction solution (3 mL) was removed every 5 min and filtered through the filter head of a disposable PES filter with a diameter of 0.45 μm. The supernatant was then collected for UV analysis. The tetracycline concentration was monitored by measuring the absorption intensity of the maximum absorption wavelength (λ = 357 nm) with a Hitachi UV-visible spectrophotometer (UH-5300, Tokyo, Japan). Pollutant decomposition efficiency (η) [30] is calculated as follows:$$\eta =(1-\frac{C}{C_{0}})\times 100\%$$

The concentration of pollutants after adsorption equilibrium is represented by *C*_0_, where *C* is the concentration of pollutants during irradiation. In order to evaluate the stability and reusability of **Zn-MOF**, cyclic photodegradation experiments were performed, and followed the same process as above. At the end of each cycle, the catalysts were collected via centrifugation, washed with deionized water and anhydrous ethanol, and dried under the same conditions for use in the next antibiotic degradation.

### 3.5. Photoelectric Response Measurement

In order to investigate the photoelectric ability of **Zn-MOF**, the photoelectric properties of the material were evaluated. The tests were performed on a CHI 660E electrochemical analyzer. In the standard three-electrode system, **Zn-MOF** powder-modified indium tin oxide (ITO) was used as the working electrode. The electrode coating area was 1.0 cm^2^. A platinum wire electrode was used as the auxiliary electrode, Ag/AgCl was used as the reference electrode and 0.5 M Na_2_SO_4_ aqueous solution was used as the electrolyte. The system was carried out in a quartz glass reactor of about 50 cm^3^ and used a 300 W xenon lamp as a light source.

## 4. Conclusions

In summary, the multifunctional nature of **Zn-MOF** has been explored with regard to antibacterial properties, photocatalytic tetracycline degradation and photoelectric activity. **Zn-MOF** has an inhibitory effect on *Escherichia coli* and *Staphylococcus aureus*, the zinc ions in which can disturb the bacterial environmental metal balance and destroy the ion channels in the membrane. For tetracycline, **Zn-MOF** exhibits recyclable photocatalytic degradation. ESR testing demonstrates the importance of the production of active substances for photodegradation. At the same time, BET, SEM, UV-vis, VB-XPS and ESR spectra jointly explained the mechanism of photocatalytic degradation of tetracycline. Finally, the photoelectronic measurement results show that the **Zn-MOF**-modified electrode can produce higher photocurrent density under negative voltage response, and the photocurrent with adjustable frequency can be obtained by changing the lighting period of the switch and maintaining a certain intensity. This study provides a new perspective for exploring the development of MOF materials in antimicrobial, degradation and photoelectric directions.

## Figures and Tables

**Figure 1 molecules-28-06662-f001:**
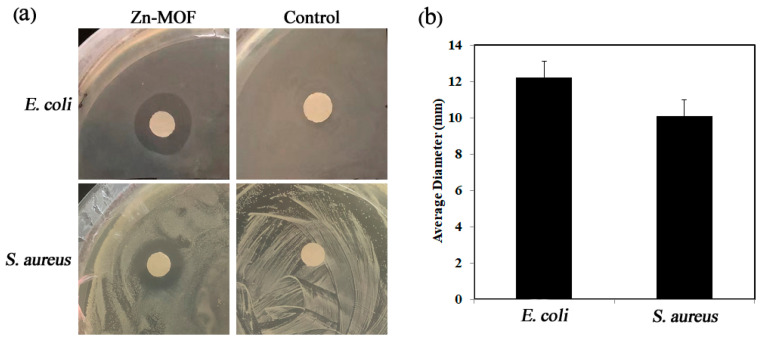
(**a**) Effect of **Zn-MOF** on bacteriostasis of *Escherichia coli* and *Staphylococcus aureus.* (**b**) Average diameter of **Zn-MOF** inhibition zone for two kinds of bacteria (mm).

**Figure 2 molecules-28-06662-f002:**
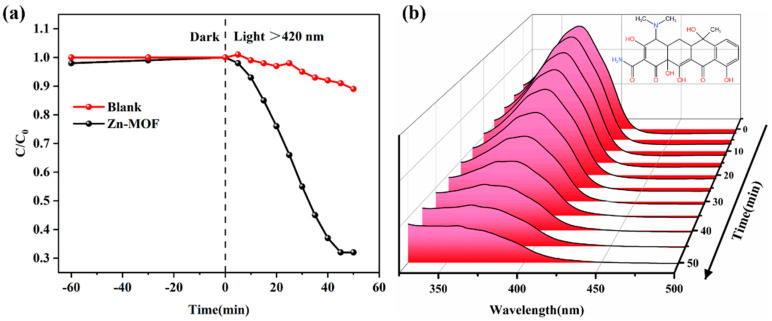
(**a**) Photodegradation of tetracycline with matte catalyst under visible light. (**b**) UV absorption monitoring of tetracycline concentration with light time degradation process.

**Figure 3 molecules-28-06662-f003:**
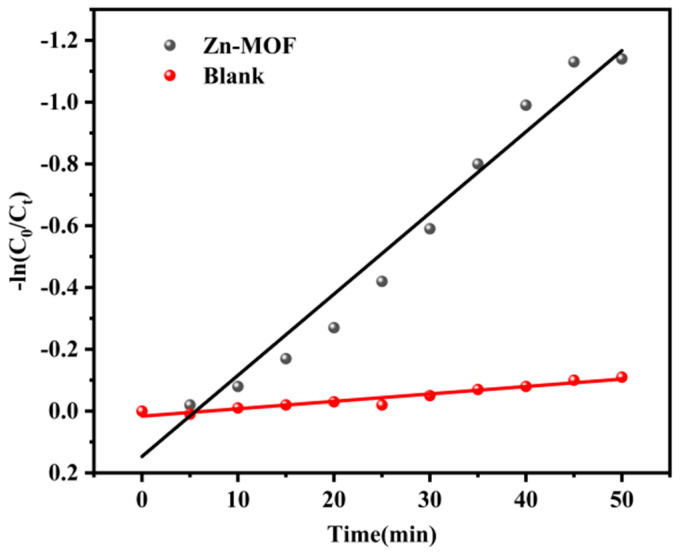
Kinetic rate constants of tetracycline degradation in the presence of matte catalysts.

**Figure 4 molecules-28-06662-f004:**
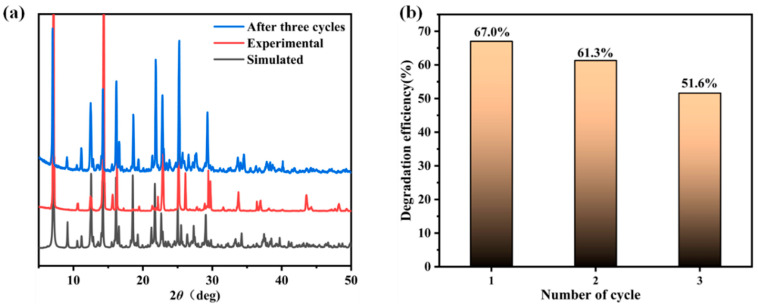
(**a**) PXRD pattern. (**b**) Recycling performance of **Zn-MOF-**degraded tetracycline.

**Figure 5 molecules-28-06662-f005:**
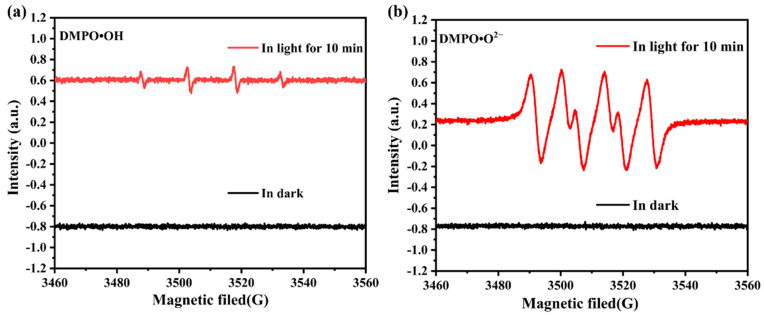
ESR spectra of (**a**) DMPO∙OH, (**b**) DMPO∙O_2_^−^.

**Figure 6 molecules-28-06662-f006:**
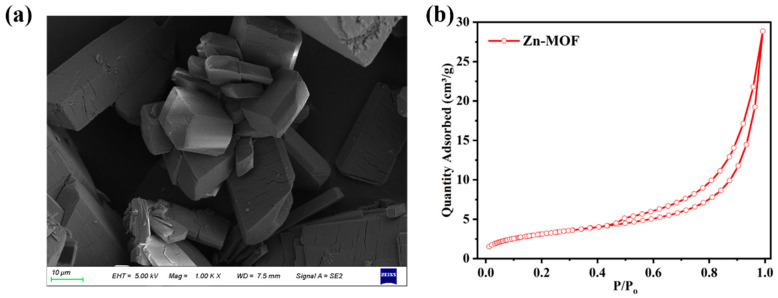
(**a**) SEM image of **Zn-MOF.** (**b**) Adsorption/desorption isotherm of **Zn-MOF**.

**Figure 7 molecules-28-06662-f007:**
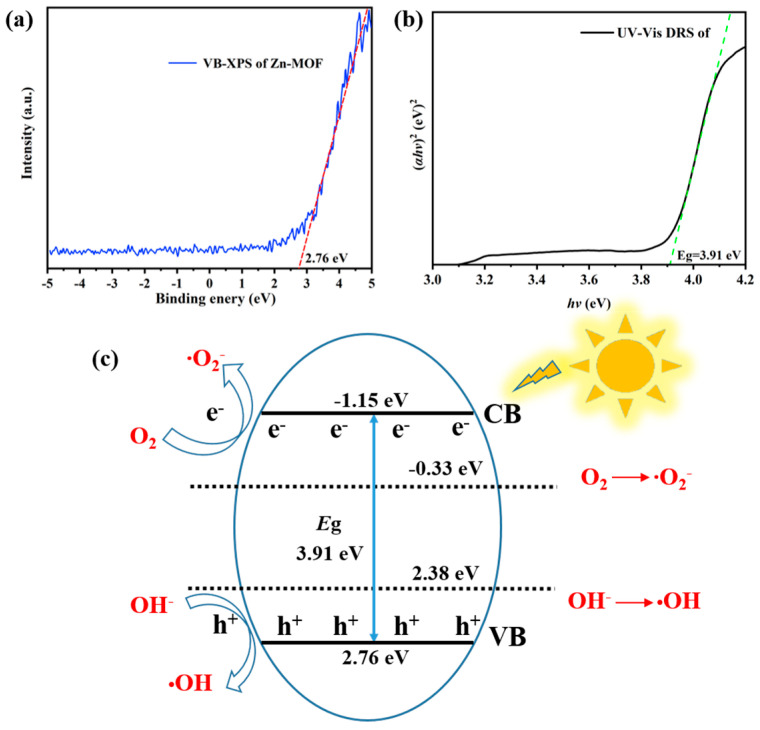
(**a**) VB-XPS spectra. (**b**) UV-vis DRS diagram. (**c**) Possible mechanism of tetracycline degradation.

**Figure 8 molecules-28-06662-f008:**
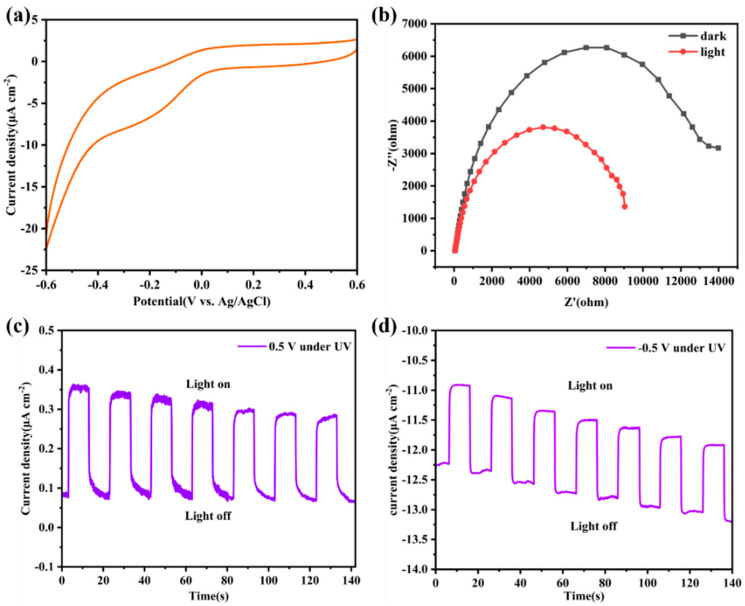
(**a**) Cyclic voltammetry of **Zn-MOF**. (**b**) Electrochemical impedance spectra (EIS) of **Zn-MOF** under dark irradiation. (**c**,**d**) Transient current density time curves of **Zn-MOF** under different bias potentials.

## Data Availability

Not applicable.

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
