# Peer review of "Highly Water-Stable Zinc Based Metal–Organic Framework: Antibacterial, Photocatalytic Degradation and Photoelectric Responses"

_molecules, 2023, doi:10.3390/molecules28186662_

Round 1

Reviewer 1 Report

Comments and Suggestions for Authors

In the current investigation, the researchers have proficiently synthesized a water-stable metal-organic framework denoted as [Zn(L)2(bpa)(H2O)2]·2H2O (referred to as Zn-MOF) through a meticulously executed solvothermal method. This synthesis involved the strategic utilization of 5-(2-cyanophenoxy)isophthalic acid (H2L), 1,2-bis(4-pyridine)-ethane (bpa), and Zn(Ac)2·2H2O. Remarkably, the study's findings unveiled the compelling antibacterial potential of Zn-MOF. The inhibitory effects were quantified by evaluating the average diameter of the inhibition zones against both Escherichia coli and Staphylococcus aureus. Notably, the average inhibition zone diameters were determined to be 12.22 mm and 10.10 mm for Escherichia coli and Staphylococcus aureus, respectively. This paper's intriguing content makes it a valuable addition to the journal's readership, particularly upon the implementation of some minor refinements:

It is recommended that the authors clearly articulate the rationale behind undertaking this study right from the outset, ideally within the opening sentence of the Abstract section.

The paper would be further enhanced by refraining from employing excessive bulk references, thus promoting a more streamlined reading experience.

The assertion pertaining to tetracycline as a prevalent environmental antibiotic necessitates proper citation for contextual accuracy.

In order to bolster the robustness of the presented findings in Figure 4, it is advised to corroborate these outcomes with prior research that bears relevance to the subject matter.

To ensure scholarly rigor, all equations introduced within the paper ought to be substantiated by appropriate references.

Within the Conclusion section, the authors are encouraged to expound comprehensively on the forthcoming trajectory of this research, elucidating the avenues for future exploration and development.

By adhering to these recommendations, the authors have the opportunity to refine their already insightful work, thereby delivering an even more impactful contribution to the scientific community.

Reviewer 2 Report

Comments and Suggestions for Authors

Dear Editor

The current manuscript presented by Yuan et al., explains an interesting work in application of a Zn-based MOF for photodegradation, photoelectrical, and antibacterial processes. Although the results seem reasonable and they have been evaluated and interpreted properly, there are some concerns that should be accurately addressed by the authors before reconsideration of manuscript for publication.

1.      There are some abbreviated words that must be defined at the first time in the manuscript. For instance, ERS (page 2), DMPO (page 4), DMSO (page 6), and …

2.      All characterization instruments used during the experiments must be explained and introduced. For instance, what type of ERS spectroscopy instrument has been employed? What about XRD?, …

3.       Some previous works have proved that the BET surface area also plays a key role in photocatalytic properties of MOFs. The authors must analyze BET surface area for their own sample

4.      It’s suggested to add SEM pictures to see the morphology of the MOF

5.      As Fig.4a shows, a significant peak around 9 degree has disappeared in the as-prepared sample; however, it can be observed again after 3 cycles of photodegradation. Why this peak is absent in the fresh sample while it is obvious in the patterns of simulation one and 3-time cycled sample

6.      The explanations regarding the performance of the MOF toward tetracycline degradation is not sufficient. What mechanism is happening during the degradation process? What are the role of OH and O2 radicals?

7.      Conclusion must be revised by adding more quantitative results

Reviewer 3 Report

Comments and Suggestions for Authors

In this paper, Zn-MOF was synthesized and applied in antibacterial purposes and tetracycline degradation. The work content of this paper is not sufficient, and the article has obvious mistakes. Some problems are listed as follows. It is recommended to make a major revision before acceptance.

1.Why did the authors choose 0.5 V and -0.5 V bias to test the photoelectric response? Please explain the reason. It suggest to test the photoelectric response at a range of gradient bias to determine the optimal bias.

2.“ESR”, “DMPO” didn't have a full name when it firstly appeared. Please check the abbreviations of professional terms carefully and correct them.

3.“using a 300 W xenon lamp equipped with a 420 nm filter to simulate the degradation of tetracycline.” Please carefully check whether the above information matches Fig. 2a.

4. Tetracycline degradation tested only once? Please test at least three times and add the error bar.

5. “the platinum wire electrode was used as the reference electrode, Ag/AgCl was used as the reference electrode”. The platinum wire electrode is usually used as the counter electrode rather than a reference electrode.

6. “Finally, the photoelectronic measurement results show that the Zn-MOF modified electrode can produce a higher photocurrent density, and the frequency-adjustable AC current can be obtained by changing the on-off-on illumination period.” In the conclusion, the author uses the comparative “higher”, but there is no comparison in the paper. In addition, alternating current refers to the current direction changes periodically, however, the current direction of the photoelectric response is always the same. That description is not correct.

7. The experimental content is not sufficient, so it is suggested to further characterize the properties and structure of the Zn-MOF. Such as TEM, SEM, UV-vis, bandgap structure, free radical trapping, carrier transfer mechanism and so on.

Comments on the Quality of English Language

Some obvious mistake can be found in the text. Please carefully modify the language.

Reviewer 4 Report

Comments and Suggestions for Authors

The paper by Ying Zhao et al. describes some of the properties of Zn-MOF, which are promising materials for practical applications. The following remarks to the work have arisen.

First, the abstract and the introduction talk about obtaining a new Zn-MOF: "Based on these questions, a novel compound with water stability, antibacterial activity and photocatalytic degradation were developed (Zn-MOF)". However, this MOF was obtained in [28]. The novelty of the study in this case is significantly reduced.

The studied properties of MOF do not allow comparison with many other similar substances. There are a lot of general phrases about MOF in the introduction, however, it is not clear why this particular MOF is of interest to researchers?

The authors claim that the ligands themselves exhibit biological activity. However, no data are given on this matter.

The main advantages of MOF are the presence of a porous structures. In this study, this does not play any role.

The relationship of the structure of MOF with properties was not specified.

Oxygen - AcO was omitted in the spelling of zinc acetate.

Based on the low novelty and poor justification of the obtained characteristics for the already known MOF, I tend to reject this work for publication in Molecules.

Round 2

Reviewer 2 Report

Comments and Suggestions for Authors

Dear Editor

The authors have diligently addressed all the comments from the reviewers. Nevertheless, there remains a single comment that necessitates their further consideration. Specifically, on page 5, the authors assert that "According to previous reports, the large specific surface area and pore volume provide favorable conditions for molecular diffusion and entry into the active site of the reactant, which reveals an important reason for the high photocatalytic activity" however, their reported surface area is not high at all. It is only 11 m2/g which is very low for MOF materials. 

The authors must explain their results in more detail to address this point.

Reviewer 3 Report

Comments and Suggestions for Authors

Now, the manuscript can be accepted.

Author Response

Thank you very much for giving us the opportunity to revise our manuscript, we reviewed and modified the full paper, and checked the order and correlation of the literature. The revised points in the manuscript are highlighted in a yellow background. 

Reviewer 4 Report

Comments and Suggestions for Authors

The authors significantly improved the manuscript and addressed most of the reviewer's comments compared to its first version. Thus, I recommend the article to publish in its current form.

Author Response

Thank you very much for taking the time to review this manuscript.  We reviewed the full paper, modified and checked the order and correlation of the literature. The revised points in the manuscript are highlighted in a yellow background.